# Mental toll on working women during the COVID-19 pandemic: An exploratory study using Reddit data

Chengyue Huang[1], Anindita Bandyopadhyay[1], Weiguo Fan[1], Aaron Miller[2], Stephanie Gilbertson-White[3]*

1 Department of Business Analytics, University of Iowa, Iowa City, Iowa, United States of America, 2 Department of Internal Medicine, Roy J. and Lucille A. Carver College of Medicine, University of Iowa, Iowa City, Iowa, United States of America, 3 College of Nursing, University of Iowa, Iowa City, Iowa, United States of America

* stephanie-gilbertson-white@uiowa.edu

**Data Availability Statement:** All relevant data are in the Supporting Information files.

**Funding:** The author(s) received no specific funding for this work.

## Abstract

COVID-19 has led to an unprecedented surge in unemployment associated with increased anxiety, stress, and loneliness impacting the well-being of various groups of people (based on gender and age). Given the increased unemployment rate, this study intends to understand if the different dimensions of well-being change across age and gender. By quantifying sentiment, stress, and loneliness with natural language processing tools and one-way, between-group multivariate analysis of variance (MANOVA) using Reddit data, we assessed the differences in well-being characteristics for age groups and gender. We see a noticeable increase in the number of mental health-related subreddits for younger women since March 2020 and the trigger words used by them indicate poor mental health caused by relationship and career challenges posed by the pandemic. The MANOVA results show that women under 30 have significantly (p = 0.05) higher negative sentiment, stress, and loneliness levels than other age and gender groups. The results suggest that younger women express their vulnerability on social media more strongly than older women or men. The huge disruption of job routines caused by COVID-19 alongside inadequate relief and benefit programs has wrecked the economy and forced millions of women and families to the edge of bankruptcy. Women had to choose between being home managers and financial providers due to the countrywide shutdown of schools and day-cares. These findings open opportunities to reconsider how policy supports women's responsibilities.

## Introduction

### Background

The pandemic has had a significant effect on unemployment in the United States. The unemployment rate reached its peak, which was the highest level since data collection began in 1948

**Competing interests:** The authors have declared that no competing interests exist.

(14.8% in April 2020), and then fell to a still-elevated level (6.7%) in December 2020 [1]. The effect of unemployment varies heavily across age and gender.

Young workers and women experienced relatively high unemployment rates during the pandemic [1]. Since the 1980s, the gender unemployment gap has been significantly narrowing but the COVID-19 crisis brought back gender inequalities [2]. While men and women had a roughly equal unemployment rate pre-pandemic, a Current Population Survey conducted by the US Census Bureau and the US Bureau of Labor Statistics shows that the unemployment for women rose to 15.8% (2% above that of men).

In occupations where working from home is possible, the attrition and lay-off rates are much smaller [3]. Especially among professionals that cannot work from home, the number of employed people fell by 15% from February to April, and the unemployment rate rose by 9% [3]. Conversely, among jobs where working from home is possible, the number of employed people fell by 7% over the same period, and the unemployment rate rose by 5% [3]. Along with spiraling unemployment, according to a Statista report, a steep rise in social media activity was seen post-COVID-19 in the US (32% increase in March 2020 alone). COVID-19 (leading to social distancing) has shown an increase in the number of social media posts [4] of unfavorable feelings and harrowing personal stories [5]. These social media posts can be related to the negative experience or psychological distress caused by unemployment due to the COVID-19 pandemic [6].

The world has seen a profound change in communication and interaction since the rapid development of social networking sites. 90% of adults in the United States use at least one social media platform [7]. Social media has become a standard resource to understand and analyze the challenges of mental health issues [8]. The prevalent methods of characterizing and estimating mental health conditions are not only expensive but also significantly time-consuming. On the contrary, social media posts supplement traditional methods while also providing potential real-time information on a massive scale. Over the past decade, social media has been the platform to co-create, share, and exchange content (views, multimedia, message, etc.) Several reports have shown that people with mental illnesses, including psychotic disorders, anxiety, or depression, use social media to express their concerns and seek a common ground for information and help [9].

### Related work

**Social media analytics.**   Data mining and information retrieval from social media using customized computational functions help opinion mining [10] and sentiment analysis [11–13]. With the help of computerized linguistic techniques, studies show the importance of text mining and how it is now possible to convert unstructured data into a structured form to obtain meaningful insights [14–18]. The prevalence of social media entails the usage of social media analytics, the stages of the analytics process, the most frequent social media analytic techniques, and the ways analytics adds value to public health and mental well-being [19, 20].

**Relevance of gender and age.**   Many social and cultural issues, such as social role expectations, discrimination, and violence, play a role in mental health and wellness. We cannot assume that distinctions are solely biological, or that they are solely cultural. Studies show that sex and gender play a significant role in the etiology, neurology, and therapy of mental diseases [21]. Hormones like estrogen and progesterone are known to have a major impact on mood, stress, and cognition. Studies have determined how these hormones may influence mental health factors ranging from the development of fear and anxiety to the risk of drug and alcohol use [22]. Gender-specific characteristics can majorly influence how treatment services are implemented [23].

Users' gender and age are often missing or falsified on social media postings, and yet they are critical for many empirical analyses. Consequently, several attempts have been made to predict social media user gender and age from user postings automatically. For example, Sap, Maarten, et al. [24] finds that the most recent 100 posts of a user can be used along with a predictive lexicon (words and weights) to predict user age and gender with good accuracy. The lexicon was built based on word usage analytics using regression and classification models with Facebook, blogs, and Twitter data along with associated demographic labels.

**Mental health assessment.** Significant research has been done in calculating and identifying stress levels using text obtained from social media (tweets, comments, posts, etc.). For example, many researchers have used machine learning and deep learning approaches based on semantic analysis of social media texts from Twitter, Reddit, and Facebook [25].

Recent studies on detecting mental health issues due to the novel COVID-19 have used VADER (Valence Aware Dictionary and sentiment Reasoner) and LIWC (Linguistic Inquiry and Word Count) extensively. Researchers have used VADER and LIWC to get sentiment scores and other well-being measures to understand the change in sentiment over time and with news regarding the pandemic [26–28]. Several other recent studies have used topic modeling and sentiment analysis to find the reasons behind mental health deterioration during the pandemic [29–32].

Short texts like Reddit posts and tweets are challenging analyzing and determine sentiment as they have a restriction on the number of characters which leads to the use of acronyms, slangs, and misspellings by users [33]. The SentiStrength [34] and TensiStrength [35] algorithms were developed by Thelwall and colleagues to measure the intensity of the positive and negative words and the intensity of stress and relaxation using a lexical approach. SentiStrength was used by Constantin Orăsan to detect aggression in social media texts [36]. This paper proved that SentiStrength performs well compared to other complicated algorithms. It was also used by Durahim et al. to calculate the Gross National Happiness of Turkey using Twitter data [37]. The output of SentiStrength was compared to the happiness survey results published by the Turkish Statistical Institute, and they were similar when the whole country was considered.

Longitudinal Twitter data across 3 case studies was used to examine the impact of violence near or on college campuses [38]. Using a computerized text analysis method (Linguistic Inquiry and Word Count, LIWC), the Tweets dataset was annotated for the presence of event-related negative emotion words.

LIWC was used to quantify mental health illness (post-traumatic stress disorder (PTSD), depression, bipolar disorder, and seasonal affective disorder (SAD)) using Twitter data [39] to demonstrate how rigorous application of simple natural language processing methods can yield insight into these mental health disorders and show linguistic signals in social media relevant to mental health.

A team developed a "social media depression index" [40] using crowd-sourced Twitter data of users diagnosed with clinical depression. This index can be utilized to measure the magnitude of depression in individuals. In addition, the same team investigates how tweets can be used as a predictor of postpartum depression [41]. They utilized measures such as social media engagement, tweet emotions, ego networks, and linguistic styles.

Another study [42] employed Twitter data to predict the onset of depression before it is reported. They first used crowd-sourced Twitter data from individuals diagnosed with clinical depression to calibrate measures from their tweets. Important measures that the team utilized were social activity, ego networks, relational and medical concerns, and religious involvement. A lexicon to detect an individual's loneliness expressions on social media is also developed in 2019 [43].

**Age- and gender-based impact on mental health.** A March 2020 poll showed the basic difference in the COVID-19 impact on men and women, where women were more likely to report feeling worried and burdened with the economic and health concerns of their families [44]. The poll shows that 57% of mothers (compared to 32% fathers) with children under the age of 18 have reported deterioration of their mental health due to the pandemic.

Austrian research [45] showed that the impact of lockdown and the pandemic, in general, was most severe for younger adults (<35 years of age), women, unemployed, and low income. The results were based on an online survey conducted after four weeks of lockdown in Austria. They performed an analysis of variance (ANOVAs) and Bonferroni-corrected post-hoc tests to compare the difference in the impact of mental health indicators (quality of life (WHO-QOL BREF), well-being (WHO-5), depression (PHQ-9), anxiety (GAD-7), stress (PSS-10), and sleep quality (ISI)) based on age, gender, employment status, and net income. Another similar survey study conducted in Italy [46] associated women with a high prevalence of sleep disturbances during the lockdown.

A longitudinal study from the Netherlands [47] showed that the pandemic had different mental health impacts based on gender—women experienced more depression and men experienced more anxiety. This was a population-based cohort study, and they used several fixed-effects models to predict the number of depression and anxiety symptoms (Poisson fixed-effects models), and occurrence of major depressive disorders, and occurrence of generalized anxiety disorders (Probit model) given gender, age, level of education, and income. One of the key results stated that women in low-income categories have had significantly more depressive symptoms and disorders than men during the lockdown. They also observed married women experienced significantly worse depressive symptoms than men.

A study in Spain showed that in a sample of 551 adults (18–25 years), 130 of them showed some level of depression (mild to extremely severe) [48]. Another cross-sectional online survey study from Spain showed that the COVID-19 lockdowns and confinements impacted the younger adults more negatively than any other age group [49].

A 2000 study in Australia used the 1995 National Health Survey and the 1997 National Survey of Mental Health and Well-being of Adults data sets to predict three indicators of mental health and well-being (psychological well-being, diagnoses of mental disorders, and suicidal thoughts, plans, and failed attempts) [50]. They used Tobit models and concluded that unemployment is a significant factor in measuring these indicators and that the unemployed people exhibited poorer mental health than the employed.

Another group studied the role of gender on unemployment and mental well-being in Sweden and Ireland [51]. They used data from long-term unemployment project (LUP) for Sweden and Living in Ireland panel survey (LII) for Ireland. Using mixed effects models, they show that Sweden, like most other countries, had women suffering more from psychological distress, anxiety, and sleeping disorders due to unemployment than men. Ireland, on the other hand, is one of the very few countries where there are no significant gender differences in mental health indicators based on unemployment.

## Objective

Prior research shows that most mental health studies are done through surveys, which are both time and resource intensive. There is a gap of research on identifying mental health issues among different age and gender groups due to unemployment during the pandemic using social media data.

In this paper, we focus on understanding and assessing the individuals' well-being during the pandemic and document the impacts they experience in terms of psychological and mental

health using freely available real-time social media data which captures key demographics instead of using time-consuming and costly survey-based data. We aim to identify if mental health issues were heightened due to age- and gender-based inequity in terms of unemployment during the pandemic.

## Materials and methods

### Data

Our data set comes from Reddit, one of the most popular sites in the US. As per a study by M. Jamnik et al. [52], it is validated that, unlike other paid sources, Reddit allows researchers to gather data and global responses of redditors. Reddit also organizes its data by subreddits [53, 54] which helps in filtering and targeting required topics easily. We use publicly available secondary data from Reddit and do not collect personal information during the text mining process. Moreover, as we do not interview or survey redditors whose posts are collected, there is no IRB (Institutional Review Board) approval or participant consent required. The collection and analysis method of this data complied with the terms and conditions of Reddit.

Apart from these, the two major advantages of Reddit over Facebook or Twitter are its redditor anonymity and freedom to write without word/character length limitations [55–58] Due to these advantages, redditors can freely discuss mental health issues and seek help from one another [59].

Since we are interested in the people that are unemployed, we use the unemployment subreddit to collect our data. We collected posts from the unemployment subreddit for members of the subreddit who have lost their jobs due to the pandemic (i.e., from January 2020 to December 2020).

Our age and gender prediction results show that about 71% of unemployed redditors in our dataset are men. The proportion of unemployed redditors under 30 years is 59.12% and over 30 is 40.88%. According to Pew Research's report about The Demographics of Reddit in 2016, people who use Reddit skew man (71%). They also found that the redditor base was 64% between the ages of 18 and 29, and another 29% were between the ages of 30 and 49. Only 6% of redditors were found to be between the ages of 50 and 64, and just a 1% was 65 or older. Our prediction results are remarkably like this report.

### Data selection

We used 1:1 matching pair design to make sure each pair is matched on gender and age. In each group, there is equal representation of both gender and age groups. Data sampling was repeated three times to avoid selection bias. Finally, we had an unbiased sample of 1,500 redditors in each of the four focus groups: women under 30, women over 30, men under 30, and men over 30.

### Methods

We studied the difference in three dimensions of well-being (i.e., stress, sentiment, and loneliness) during the pandemic among the four aforementioned groups of people (women under 30, women over 30, men under 30, and men over 30).

To quantify redditors' well-being, we measure the sentiment score and stress score in each post as well as loneliness score for each redditor. Example posts of showing positive and negative sentiment, and stress and relax are shown in Table 1.

Sentiment and stress/relax are measured for every Reddit post of each user. But loneliness is evaluated at a redditor level and not at a post level. So, no example for loneliness is shown here.

**Table 1. Example of posts displaying extremes of stress and sentiment.**

| | Score | Example posts |
|---|---|---|
| **Low stress** | 4.00 | Got a line cook job after 6 months of unemployment couldnt be more relieved!!! |
| **High stress** | -4.00 | I am extremely broke like to the point where it's scary. But I feel like I need to start completely fresh. The world is scary right now and I feel like going backwards is never the right way to go. . . I am so stressed out from all of this and do not kno how to get it off my mind. |
| **Positive Sentiment** | 4.00 | These are adorable and I want them! |
| **Negative Sentiment** | -4.00 | I am also worried they will be able to come into my apartment unannounced now. I am really scared what is going to happen to me. I have nowhere else to go or any money at all. |

**Sentiment.** We use SentiStrength [34] which is a lexicon and rule-based sentiment analysis tool that is specifically attuned to sentiments expressed in short social media texts.

**Stress.** To measure stress, TensiStrength [35] is used. It uses a lexical approach and a set of rules to detect direct and indirect expressions of stress or relaxation.

A post gets a separate positive (1 to 5) and negative (-1 to -5) score, and their summation gives the overall score. Thus, the overall score ranges from -4 to 4. This method is true for calculating both sentiment and stress scores.

**Loneliness.** We applied a pre-trained data-driven machine learning model to detect an individual's loneliness expression in posts [43]. High value means an elevated level of loneliness.

**MANOVA test.** A one-way, between-group MANOVA [60] is conducted to examine difference in well-being measures between four different gender and age groups. There are three dependent variables: sentiment, stress, and loneliness.

We use Wilks' lambda test statistic to test if there are differences in age and gender group means for the combination of dependent variables. To determine how the dependent variables differ for the independent gender and age groups we run multiple (three in our case) univariate ANOVAs (analysis of variance) [61, 62]. Finally, we follow-up the significant ANOVA tests with Tukey's HSD (honestly significant difference) post-hoc tests [63] to check if the well-being measures are significantly different between each gender and age group.

## Results

As per our analysis, women are more expressive about their experiences of affected by unemployment during the pandemic. The number of posts in mental health subreddits increased markedly for women irrespective of their age from April 2020 (Fig 1). Upon deepening our research, we can see a striking difference in the number of posts under mental health subreddit between the two age groups irrespective of gender (Fig 2).

From the word cloud (Fig 3), we can identify the most common areas of discussion on Reddit for each gender and age group. While men mainly discuss video games ("player", "catacomb", "warband", "game", "switch", etc.), women chew over relationships ("husband", "parent"), mental health problems ("anxiety", "pain", "therapist", "abuse"), careers ("school", "college", "work"), and financial instability ("benefit", "house", "claim", "fund", "account"). This shows how the pandemic has intensified the already existing gender disparity.

For the MANOVA tests, a 5% level of significance (alpha, $\alpha = 0.05$) was used to reduce the risk of type 1 errors [64]. The multivariate test using the Wilks' Lambda test statistic shows that there is a significant main effect of gender and age group on well-being measures with p-value $<0.001$.

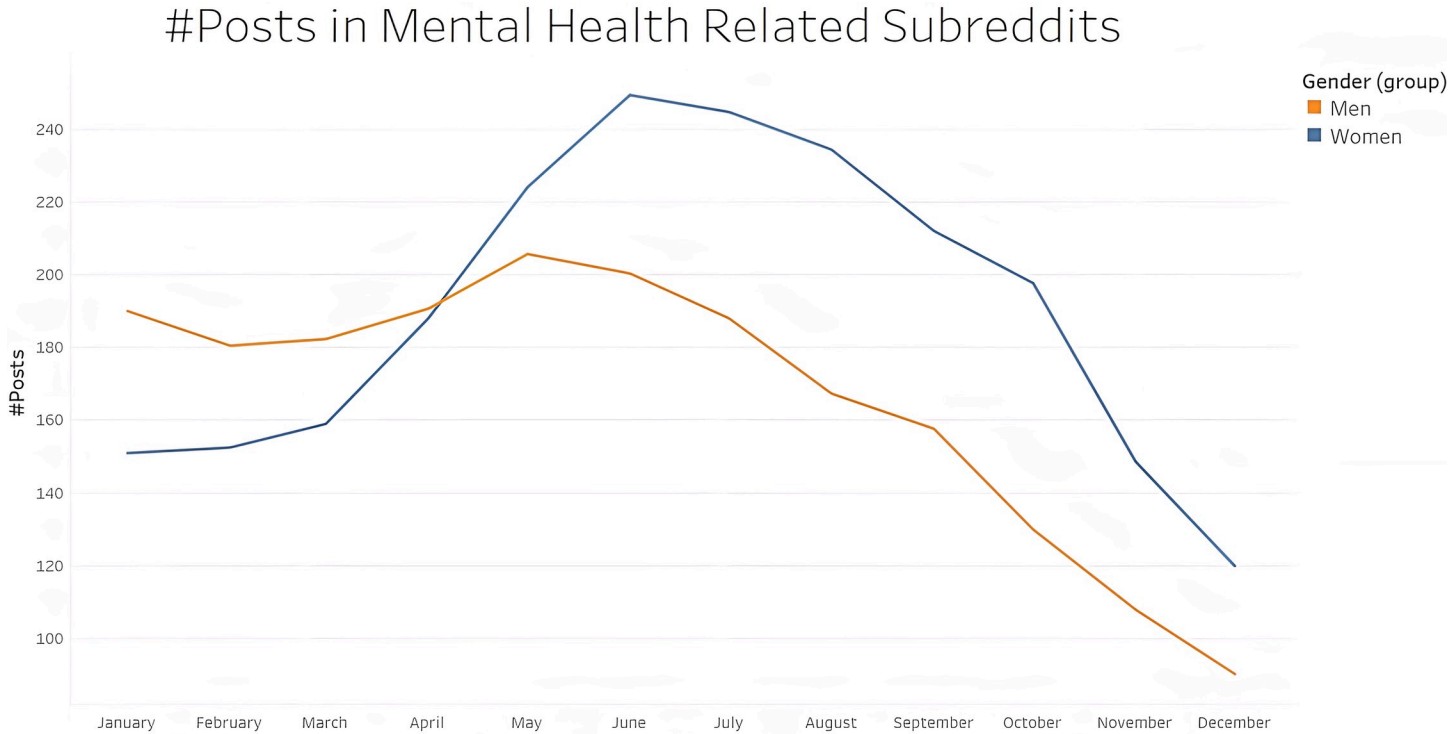

**Fig 1. Change in number of mental-health related posts in 2020 based on gender.** The orange line represents all unemployed women redditors and the blue represents all unemployed men redditors. Starting from March 2020, the number of mental health related subreddits increased noticeably for women.

The p-value (column "significance" in Table 2) for each well-being measure is <0.001 showing that there is a significant effect of gender and age groups on all the three well-being measures. We used a Bonferroni corrected alpha to account for the multiple ANOVA tests, which was assessed by dividing the conservative alpha of 0.05 by the number of dependent variables (Bonferroni corrected $\alpha = 0.05/3 = 0.0167$). We can also see from the descriptive statistics table (Table 3) that women below 30 years of age have the lowest mean sentiment (the highest negative sentiment) and stress scores (the highest stress), and highest mean loneliness score compared to other gender and age groups. This means that younger women experienced higher negative sentiment, stress, and loneliness than all other gender and age groups.

The results of MANOVA (Table 4) indicate the significant changes in sentiment, stress, and loneliness levels for separate groups of people. For sentiment, we see that the mean sentiment score is statistically significantly different between women below 30 and men of all ages. For stress, we find equivalent results as sentiment with the addition of the mean stress score being statistically significantly different between women and men above 30 years of age. For loneliness as well we see equivalent results as sentiment with the addition of the mean loneliness score is statistically significantly different between women below and above 30 years of age.

## Discussion

The key findings of the study were that women have been more vocal about their experiences with unemployment throughout the pandemic and while men used social media (Reddit) to talk about video games, women were more interested in talking about employment, mental health issues, and financial security, revealing a widening gender gap. The MANOVA tests

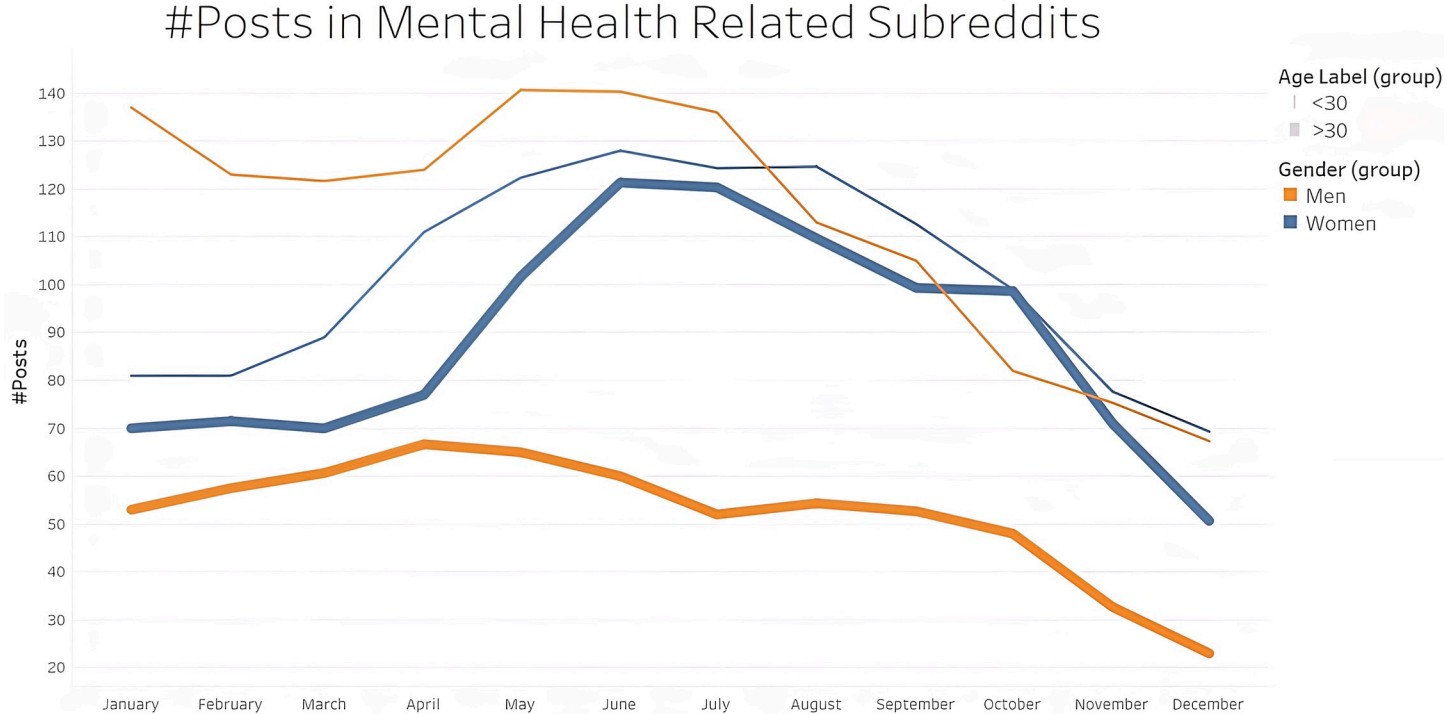

**Fig 2. Change in number of mental-health related posts in 2020 based on age and gender.** The orange and blue lines represent women and men respectively. The thin lines are for people below 30 years of age and the thick lines are for people above 30 years of age. Mental health related posts are far more by younger people than older, irrespective of gender.

also revealed that women (particularly those under 30) experience much higher negative sentiment, stress, and loneliness than men of similar ages.

Based on our analysis, women younger than 30 may be the most psychologically impacted group during the pandemic. Their negative sentiment, stress, and loneliness levels are significantly higher than other groups, especially men.

The United States recorded its highest unemployment rate of 14.8% in April 2020 [1].

The rate of unemployment for women in the United States increased from 3.4% in February 2020 to 15.7% in April 2020, which was a higher spike than that for men (4.1% to 13.3% over this period) [65]. As per a policy brief by the United Nations [66], women will be the hardest hit in every aspect (health, economy, social safety, etc.) during this pandemic simply by the virtue of their gender. Indeed, Fig 1 shows that women post more frequently in the mental health subreddit. Additionally, Fig 3 demonstrates the wide range of topics where women express their concern and worry.

A response to the escalated rate of unemployment among young workers [67] is the increase in the number of social media activity and chatter, as evident from Fig 2. As per an October 2020 report by the Economic Policy Institute, the unemployment rate for young workers in the United States has increased from 8.4% to 24.4% from Spring 2019 to Spring 2020. While it is only 2.8% to 11.3% for the older age group.

Therefore, the results of this study are not surprising given the higher unemployment rate of young workers and women during the pandemic.

A report by the Center for American Progress revealed that over four times as many women as men dropped out of the labor force, with 865,000 women compared to 216,000 men

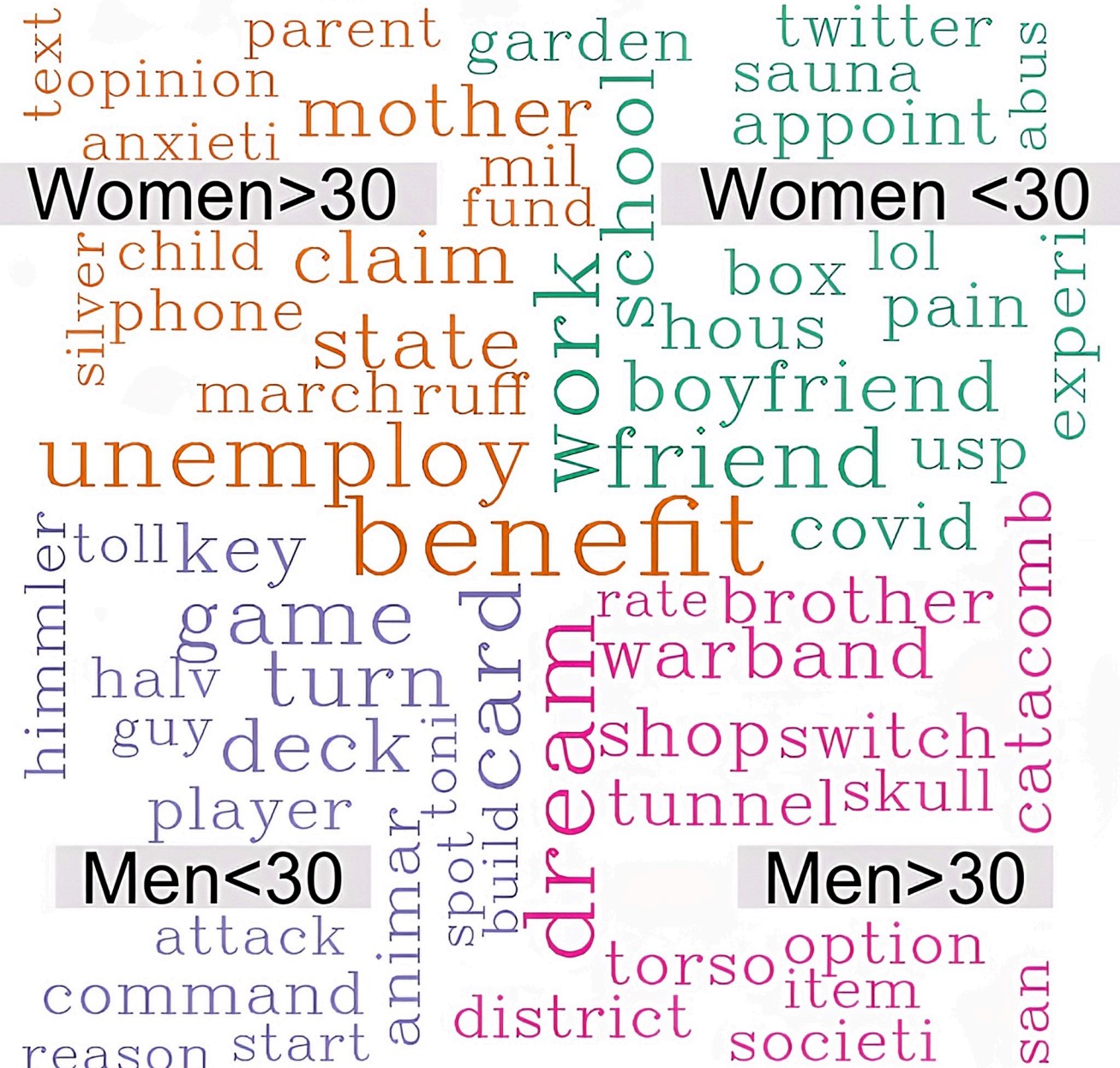

**Fig 3. Word cloud to compare the most frequent words among the four groups.**

**Table 2. Univariate ANOVA tests.**

| Dependent Variables | df | F | significance |
|---|---|---|---|
| Sentiment | 3 | 1.134 | <0.001 |
| Stress | 3 | 1.520 | <0.001 |
| Loneliness | 3 | 3.925 | <0.001 |

df–degrees of freedom, F–value of the F-statistic.

**Table 3. Descriptive statistics.**

| Dependent Variables | Labels | Mean |
|---|---|---|
| | Women <30 | -0.04 |
| Sentiment: | Women >30 | -0.00 |
| (-4 to 4) | Men <30 | 0.01 |
| | Men >30 | 0.02 |
| | Women <30 | -0.40 |
| Stress: | Women >30 | -0.37 |
| (-4 to 4) | Men <30 | -0.33 |
| | Men >30 | -0.33 |
| | Women <30 | 0.44 |
| Loneliness | Women >30 | 0.35 |
| (0 to infinity) | Men <30 | 0.35 |
| | Men >30 | 0.33 |

[68]. This confirms forecasts that the COVID-19 pandemic would have a devastating impact on women, as well as the ensuing childcare and educational issues.

According to a March 2020 NASEM survey, difficulties caused by the pandemic had a negative impact on women. Of those who responded, 28% said their workload had increased, while 25% said their productivity had deteriorated [69]. Two-thirds of those polled reported negative effects on their personal well-being, including their mental and physical health. The report further urged that woman academic scientists would profit if universities and workplaces implemented policies such as extending grants and expanding the period required for attaining a tenure—tactics that provide women more time to care for their families without jeopardizing their careers.

To address the widening gender gap resulting from the pandemic, gender-responsive policies and budgets are imperative. To mitigate the negative impacts, it is essential to provide well-designed fiscal policies to supplement incomes and employment [70]. Also, improving the social protection systems for women, like paid leaves and coverage, will make them financially stable and reduce the discontinuation of their careers [71].

## Comparison with prior work

*Factors associated with psychological distress during the coronavirus disease 2019 (COVID19) pandemic on the predominantly general population: A systematic review and meta-analysis* [72]. This paper summarizes the results of a time-consuming and costly survey-based study. It made a similar conclusion with us: women and young workers are significantly affected more in terms of well-being during the pandemic. We not only used more advanced NLP methods but also implemented temporal analysis. Our study is more cost and time-effective than surveys since we use publicly available Reddit posts that are not limited to a certain demographic population.

*Social Media Insights into US Mental Health During the COVID-19 Pandemic: Longitudinal Analysis of Twitter Data* [29]. This paper studies the negative effect of the pandemic on the general population with Twitter data. Our study focuses on the section that has undergone a transition in work culture or has lost their jobs due to this pandemic. Moreover, our study targets to identify the most vulnerable and the ones seeking support through Reddit posts. We measured not only sentiment, but also stress, loneliness, and several posts in mental health-related subreddits. The advantage of Reddit posts over tweets is that Reddit has no word/

**Table 4. Multiple comparisons using Tukey's HSD post hoc tests.**

| Dependent variables | Group I | Group J | Mean difference (I-J) |
|---|---|---|---|
| Sentiment: (-4 to 4) | Women <30 | Women >30 | -0.037 |
| | | Men <30 | -0.055* |
| | | Men >30 | -0.061* |
| | Women >30 | Men <30 | -0.017 |
| | | Men >30 | -0.024 |
| | Men <30 | Men >30 | -0.006 |
| Stress: (-4 to 4) | Women <30 | Women >30 | -0.027 |
| | | Men <30 | -0.061* |
| | | Men >30 | -0.069* |
| | Women >30 | Men <30 | -0.034 |
| | | Men >30 | -0.042* |
| | Men <30 | Men >30 | -0.008 |
| Loneliness: (0 to infinity) | Women <30 | Women >30 | 0.095* |
| | | Men <30 | 0.091* |
| | | Men >30 | 0.115* |
| | Women >30 | Men <30 | -0.003 |
| | | Men >30 | 0.020 |
| | Men <30 | Men >30 | 0.023 |

The mean difference values marked with asterisk (*) are statistically significant (p-value < α).

character limitation making it extremely helpful for redditors to freely pen down their emotions and helplessness.

## Limitations

The limitations of our work include that redditors may not be representative of all demographic segments (based on age, gender, occupation, cultural background, and family status) of the population. Social media data also poses the threat of selection bias. Regardless how widely used Reddit is, social media data might not represent the entire real-world population. As we mentioned earlier when discussing the data we used, Reddit has particular biases in its user population, especially in gender and age. To solve this problem, we should collect data from other social media sources like Twitter, which have a much broader user base. In the future, we would replicate our analysis on the Twitter data and make comparisons between users from two different social media sites. Another limitation is that currently, we have gender, age, and employment status as the only demographics of the redditors. Future investigations should extract information on location, industry, and personality as well.

## Conclusions

This study uses Reddit data instead of survey data which is both cost-effective and time efficient. Unlike other social media sites like Facebook and Twitter, Reddit enables Redditors (i.e., Reddit users) to have an honest discussion about mental health disorders while also protecting their identity and any potential social discrimination.

The results and approach of this study can be used to evaluate the psychological costs of pandemic-related prevention measures like lockdowns and closures on vulnerable groups like the young and women labor forces. Since these data are available in real-time and the posts of the high-risk redditors can be tracked over time, we can measure the indirect costs or benefits

of the pandemic and associated prevention efforts. This will also help target the sections of the population that might be most impacted and evaluate policy measures to improve social protection systems as they are rolled out. It is also advisable that health care providers may want to be attuned to the mental health impacts of unemployment & work from home.

## Supporting information

**S1 Data. This file contains the minimal data set required to reach the conclusions drawn in the manuscript.**
(CSV)

## Author Contributions

**Conceptualization:** Chengyue Huang, Anindita Bandyopadhyay, Weiguo Fan, Aaron Miller, Stephanie Gilbertson-White.

**Data curation:** Chengyue Huang.

**Formal analysis:** Chengyue Huang, Anindita Bandyopadhyay.

**Investigation:** Chengyue Huang, Anindita Bandyopadhyay.

**Methodology:** Chengyue Huang, Anindita Bandyopadhyay.

**Supervision:** Weiguo Fan, Aaron Miller, Stephanie Gilbertson-White.

**Visualization:** Chengyue Huang.

**Writing – original draft:** Chengyue Huang, Anindita Bandyopadhyay.

**Writing – review & editing:** Weiguo Fan, Aaron Miller, Stephanie Gilbertson-White.

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
