## [Decision Letter · Decision Letter 0]

10 Aug 2022

PONE-D-22-16256Mental Toll on Working Women During the COVID-19 Pandemic: An Exploratory Study Using Reddit DataPLOS ONE

Dear Dr. Gilbertson-White,

Thank you for submitting your manuscript to PLOS ONE. After careful consideration, we feel that it has merit but does not fully meet PLOS ONE’s publication criteria as it currently stands. Therefore, we invite you to submit a revised version of the manuscript that addresses the points raised during the review process.

We look forward to receiving your revised manuscript.

Kind regards,

Zhuo Chen, Ph.D.

Academic Editor

PLOS ONE

Journal Requirements:

2. In your Methods section, please include additional information about your dataset and ensure that you have included a statement specifying whether the collection and analysis method complied with the terms and conditions for the source of the data.

Reviewers' comments:

Reviewer's Responses to Questions

**Comments to the Author**

1. Is the manuscript technically sound, and do the data support the conclusions?

Reviewer #1: Partly

Reviewer #2: Partly

Reviewer #3: Yes

2. Has the statistical analysis been performed appropriately and rigorously? 

Reviewer #1: Yes

Reviewer #2: Yes

Reviewer #3: Yes

3. Have the authors made all data underlying the findings in their manuscript fully available?

Reviewer #1: Yes

Reviewer #2: Yes

Reviewer #3: Yes

4. Is the manuscript presented in an intelligible fashion and written in standard English?

Reviewer #1: Yes

Reviewer #2: Yes

Reviewer #3: Yes

5. Review Comments to the Author

Reviewer #1: This paper used social media data to test whether there were significant different in people’s mental health in terms of unemployment during the COVID-19 epidemics. Nature language processing and MANOVA were adopted to test the hypotheses, and the results supported the arguments. The language is fluent with few spelling or grammar errors. The research topic is quite interesting and important.

However, there are several issues that I would like to mention:

1) One major concern I have is that the research data and results may not be able to answer all the research questions. The aim of this research was “to identify if mental health issues were heightened due to age- and gender-based inequity in terms of unemployment during the pandemic”. While the results showed that “that younger women express their vulnerability on social media more strongly than older women or men.” Men have been proved to be less likely to express their vulnerability. And even they are talking about games on social media platform, they may actually be experiencing huge stress at the same time. The authors should justify whether they measured the expressed health issues or the actual health issues.

2) One of the literature review part names “predicting user gender and age”. However, there is only one literature in that part was talking about predicting gender and age, other reviewed literatures were talking about the importance of related topic. More related literatures on “predicting user gender and age” should be reviewed or the authors could consider changing a sub-title.

3) Sharp pictures with higher resolution should be provided.

Reviewer #2: 1. Pg. 2, background – helpful to provide the 1948 unemployment rate number as a reference

2. Pg. 5, Objective – after reading this objective and the background, it seems that the authors are summarizing findings from existing studies but not addressing a clear research gap. It would be helpful if the authors could elaborate on the literature gaps or weaknesses of existing works and how this work could potentially help.

3. Pg. 5-6, data selection & comparison groups – is it a 1-to-1 match or propensity score matching? What are the three pairs? Those four groups seem naturally to be four pairs: 1) women under 30 vs. over 30; 2) men under 30 vs. over 30; 3) under 30 women vs. men; 4) over 30 women vs. men. Based on later sections, the authors seem to be using three dimensions (sentiment, stress, and loneliness), but they are still different from the definition of pairs. Elaborating on this could help remove confusion.

4. Pg. 7, results – figures have low resolutions. For example, figure 1 (original graph added as supplemental material) has a horizontal and vertical DPI of 96, which is lower than the 300-dpi standard journal requirements. I would suggest the authors regenerate those in higher resolutions to be legible. Besides, figures 1 and 2 did not provide any legends.

5. Pg. 7, results – “We used a Bonferroni corrected alpha was used to account for the multiple ANOVA tests,” – this is a grammatically wrong sentence (used appeared twice with unclear subjective). Please double-check the manuscript for such errors.

6. Pg. 7, results – “We can also see from the descriptive statistics table (Table 3)” – there are no statistical significance levels reported in Table 3, so these interpretations may not be statistically significant.

7. Pg. 9, principal results – “Unemployed and work from home redditors in 2020 are psychologically affected by the pandemic.” This statement is not supported by the results. To address this, the author could do a thematic analysis of texts mentioning “pandemic” in the reddits. A casual or association relationship could not be established because these reddits happened during the pandemic – concurrence does not mean causal relationships. Their scenarios could be heterogeneous and may not be uniformly influenced by the pandemic.

8. Pg. 10, limitations – another limitation is the digital world could not represent the entire world, both the coverage and depth of the messages. Still, a large group of people would not have regular access to the internet or prefer not to engage in social media discussions. The authors should mention the bias from the unobserved population. On the other hand, posts expressed on social media are edited versions of reality so that they could be exaggerated or distorted.

Reviewer #3: 1. In the "Data" section, the authors reported that they collected data from the unemployment subreddit and work from home subreddit. But in the next paragraph, the authors only reported about the time frame for collecting data from the unemployment subreddits. Please clarify the time frame and relevant details for collecting data from work from home subreddit.

2. In the "Data Selection" section, the authors mentioned that they sampled 1500 redditors in each of the four groups. Please add more details about how were these redditors sampled.

3. In the "Well-being Measurements" section, the authors reported that they used NLP tools to assess sentiment, stress and loneliness. How good are the validity of these tools? Please clarify if possible.

4. In general, the authors should be more explicit about their contribution to the existent literature.

6. PLOS authors have the option to publish the peer review history of their article (what does this mean?). If published, this will include your full peer review and any attached files.

Reviewer #1: No

Reviewer #2: No

Reviewer #3: No

---

## [Author Response · Author response to Decision Letter 0]

25 Sep 2022

Journal Requirements:

We have made the necessary changes.

2. In your Methods section, please include additional information about your dataset and ensure that you have included a statement specifying whether the collection and analysis method complied with the terms and conditions for the source of the data.

We have added the extra information as requested.

Done.

Done.

Done.

Done.

Comments to the Author

5. Review Comments to the Author

Reviewer #1: 

This paper used social media data to test whether there were significant different in people’s mental health in terms of unemployment during the COVID-19 epidemics. Nature language processing and MANOVA were adopted to test the hypotheses, and the results supported the arguments. The language is fluent with few spelling or grammar errors. The research topic is quite interesting and important.

However, there are several issues that I would like to mention:

1. One major concern I have is that the research data and results may not be able to answer all the research questions. The aim of this research was “to identify if mental health issues were heightened due to age- and gender-based inequity in terms of unemployment during the pandemic”. While the results showed that “that younger women express their vulnerability on social media more strongly than older women or men.” Men have been proved to be less likely to express their vulnerability. And even they are talking about games on social media platform, they may actually be experiencing huge stress at the same time. The authors should justify whether they measured the expressed health issues or the actual health issues.

We have added a clarifying statement noting that we have measured only the expressed health issues and not the actual health issues.

2. One of the literature review part names “predicting user gender and age”. However, there is only one literature in that part was talking about predicting gender and age, other reviewed literatures were talking about the importance of related topic. More related literatures on “predicting user gender and age” should be reviewed or the authors could consider changing a sub-title.

We have changed the sub-title to emphasize the relevance of gender and age.

3. Sharp pictures with higher resolution should be provided.

We have added pictures with higher resolution.

Reviewer #2: 

1. Pg. 2, background – helpful to provide the 1948 unemployment rate number as a reference

We have added the reference.

2. Pg. 5, Objective – after reading this objective and the background, it seems that the authors are summarizing findings from existing studies but not addressing a clear research gap. It would be helpful if the authors could elaborate on the literature gaps or weaknesses of existing works and how this work could potentially help.

We have added a paragraph under the “objective” section clearly stating the research gap and how our work can potentially help.

3. Pg. 5-6, data selection & comparison groups – is it a 1-to-1 match or propensity score matching? What are the three pairs? Those four groups seem naturally to be four pairs: 1) women under 30 vs. over 30; 2) men under 30 vs. over 30; 3) under 30 women vs. men; 4) over 30 women vs. men. Based on later sections, the authors seem to be using three dimensions (sentiment, stress, and loneliness), but they are still different from the definition of pairs. Elaborating on this could help remove confusion.

We apologize for the confusion. We have clarified the difference between pairs and dimensions by explicitly stating the four pairs and three dimensions used in this study.

4. Pg. 7, results – figures have low resolutions. For example, figure 1 (original graph added as supplemental material) has a horizontal and vertical DPI of 96, which is lower than the 300-dpi standard journal requirements. I would suggest the authors regenerate those in higher resolutions to be legible. Besides, figures 1 and 2 did not provide any legends.

We have added pictures with higher resolution.

5. Pg. 7, results – “We used a Bonferroni corrected alpha was used to account for the multiple ANOVA tests,” – this is a grammatically wrong sentence (used appeared twice with unclear subjective). Please double-check the manuscript for such errors.

We apologize for this error. We have made the correction.

6. Pg. 7, results – “We can also see from the descriptive statistics table (Table 3)” – there are no statistical significance levels reported in Table 3, so these interpretations may not be statistically significant.

We do not claim that it is statistically significant.

7. Pg. 9, principal results – “Unemployed and work from home redditors in 2020 are psychologically affected by the pandemic.” This statement is not supported by the results. To address this, the author could do a thematic analysis of texts mentioning “pandemic” in the reddits. A casual or association relationship could not be established because these reddits happened during the pandemic – concurrence does not mean causal relationships. Their scenarios could be heterogeneous and may not be uniformly influenced by the pandemic.

Since this is not the main focus of the paper, we have removed the sentence in question.

8. Pg. 10, limitations – another limitation is the digital world could not represent the entire world, both the coverage and depth of the messages. Still, a large group of people would not have regular access to the internet or prefer not to engage in social media discussions. The authors should mention the bias from the unobserved population. On the other hand, posts expressed on social media are edited versions of reality so that they could be exaggerated or distorted.

We have added a limitation that talks about this.

Reviewer #3: 

1. In the "Data" section, the authors reported that they collected data from the unemployment subreddit and work from home subreddit. But in the next paragraph, the authors only reported about the time frame for collecting data from the unemployment subreddits. Please clarify the time frame and relevant details for collecting data from work from home subreddit.

Apologies for the confusion. We have removed work from home users from the study and mention of it from the manuscript.

2. In the "Data Selection" section, the authors mentioned that they sampled 1500 redditors in each of the four groups. Please add more details about how were these redditors sampled.

We used 1:1 matching pair design to make sure each pair is matched on gender and age. In each group, there is equal representation of both gender and age groups. Data sampling was repeated three times to avoid selection bias. Finally, we had an unbiased sample of 1,500 redditors in each of the four focus groups: women under 30, women over 30, men under 30, and men over 30. 

3. In the "Well-being Measurements" section, the authors reported that they used NLP tools to assess sentiment, stress, and loneliness. How good are the validity of these tools? Please clarify if possible.

There are several studies mentioned in the manuscript that talk about these tools and their performances.

4. In general, the authors should be more explicit about their contribution to the existent literature.

We have added a paragraph under the “objective” section clearly stating the research gap and how our work can potentially help.

---

## [Decision Letter · Decision Letter 1]

3 Nov 2022

PONE-D-22-16256R1Mental Toll on Working Women During the COVID-19 Pandemic: An Exploratory Study Using Reddit DataPLOS ONE

Dear Dr. Gilbertson-White,

Thank you for submitting your manuscript to PLOS ONE. The reviewers have agreed that your manuscript has met the criteria for publication but there are some minor issues to be addressed. Therefore, we invite you to submit a revised version of the manuscript that addresses the points raised during the review process.

We look forward to receiving your revised manuscript.

Kind regards,

Zhuo Chen, Ph.D.

Academic Editor

PLOS ONE

Journal Requirements:

Reviewers' comments:

Reviewer's Responses to Questions

**Comments to the Author**

1. If the authors have adequately addressed your comments raised in a previous round of review and you feel that this manuscript is now acceptable for publication, you may indicate that here to bypass the “Comments to the Author” section, enter your conflict of interest statement in the “Confidential to Editor” section, and submit your "Accept" recommendation.

Reviewer #1: All comments have been addressed

Reviewer #2: All comments have been addressed

Reviewer #3: All comments have been addressed

2. Is the manuscript technically sound, and do the data support the conclusions?

Reviewer #1: Yes

Reviewer #2: Yes

Reviewer #3: Yes

3. Has the statistical analysis been performed appropriately and rigorously? 

Reviewer #1: Yes

Reviewer #2: Yes

Reviewer #3: Yes

4. Have the authors made all data underlying the findings in their manuscript fully available?

Reviewer #1: Yes

Reviewer #2: Yes

Reviewer #3: No

5. Is the manuscript presented in an intelligible fashion and written in standard English?

Reviewer #1: Yes

Reviewer #2: Yes

Reviewer #3: Yes

6. Review Comments to the Author

Reviewer #1: The authors have addressed majority questions raised by the Editor and reviewers. It would be more helpful if the authors could provide the location and brief summary for some of the modifications in the response letter in the future.

I am happy to recommend this manuscript for publication after more proof reading and format correction.

Reviewer #2: 1. Figures still need higher resolution and legends (maybe revised figures got lost in transmission?). I will defer to the journal editor and production team on how to address this.

2. Limitations: “Social media data also poses the threat of selection bias.” This one sentence may not be sufficient. Please make sure to elaborate a bit more on the “how”.

All other comments addressed. Thanks!

Reviewer #3: (No Response)

7. PLOS authors have the option to publish the peer review history of their article (what does this mean?). If published, this will include your full peer review and any attached files.

Reviewer #1: No

Reviewer #2: No

Reviewer #3: No

---

## [Author Response · Author response to Decision Letter 1]

18 Dec 2022

Rebuttal Letter

Journal Requirements:

There was only one reference that was from a website that is longer present, so we removed it. A few of the references did not show the hyperlinks correctly or had some sort of problem due to formatting. We made changes to all of them and now all the references should be valid and easily accessible. Hope this helps with this issue raised.

Review Comments to the Author

Reviewer #1: The authors have addressed majority questions raised by the Editor and reviewers. It would be more helpful if the authors could provide the location and brief summary for some of the modifications in the response letter in the future.

I am happy to recommend this manuscript for publication after more proof reading and format correction.

Thank you for the suggestion. We will keep this in mind for the future.

Reviewer #2: 

1. Figures still need higher resolution and legends (maybe revised figures got lost in transmission?). I will defer to the journal editor and production team on how to address this.

We tried our best to address this issue and hopefully we have solved the problem of resolution this time. We have double checked the resolution and the legends in the figures.

2. Limitations: “Social media data also poses the threat of selection bias.” This one sentence may not be sufficient. Please make sure to elaborate a bit more on the “how”.

We added the following to the limitation section:

“Social media data also poses the threat of selection bias. Regardless how widely used Reddit is, social media data might not represent the entire real-world population. As we mentioned earlier when discussing the data we used, Reddit has particular biases in its user population, especially in gender and age.”

Reviewer #3: (No Response)

---

## [Editor Report · Decision Letter 2]

20 Dec 2022

Mental Toll on Working Women During the COVID-19 Pandemic: An Exploratory Study Using Reddit Data

PONE-D-22-16256R2

Dear Dr. Gilbertson-White,

I have reviewed your revised manuscript and response to the request for minor revisions. We’re pleased to inform you that your manuscript has been judged scientifically suitable for publication and will be formally accepted for publication once it meets all outstanding technical requirements.

Kind regards,

Zhuo Chen, Ph.D.

Academic Editor

PLOS ONE

---

## [Editor Report · Acceptance letter]

3 Jan 2023

PONE-D-22-16256R2 

Mental toll on working women during the COVID-19 pandemic: An exploratory study using Reddit data 

Dear Dr. Gilbertson-White:

I'm pleased to inform you that your manuscript has been deemed suitable for publication in PLOS ONE. Congratulations! Your manuscript is now with our production department. 

Kind regards, 

on behalf of

Prof. Zhuo Chen 

Academic Editor

PLOS ONE